# Hierarchical Reinforcement Learning via Advantage-Weighted Information Maximization

**Takayuki Osa**
University of Tokyo, Tokyo, Japan
RIKEN AIP, Tokyo, Japan
`osa@mfg.t.u-tokyo.ac.jp`

**Voot Tangkaratt**
RIKEN AIP, Tokyo, Japan
`voot.tangkaratt@riken.jp`

**Masashi Sugiyama**
RIKEN AIP, Tokyo, Japan
University of Tokyo, Tokyo, Japan
`sugi@k.u-tokyo.ac.jp`

## Abstract

Real-world tasks are often highly structured. Hierarchical reinforcement learning (HRL) has attracted research interest as an approach for leveraging the hierarchical structure of a given task in reinforcement learning (RL). However, identifying the hierarchical policy structure that enhances the performance of RL is not a trivial task. In this paper, we propose an HRL method that learns a latent variable of a hierarchical policy using mutual information maximization. Our approach can be interpreted as a way to learn a discrete and latent representation of the state-action space. To learn option policies that correspond to modes of the advantage function, we introduce *advantage-weighted importance sampling*. In our HRL method, the gating policy learns to select option policies based on an option-value function, and these option policies are optimized based on the deterministic policy gradient method. This framework is derived by leveraging the analogy between a monolithic policy in standard RL and a hierarchical policy in HRL by using a deterministic option policy. Experimental results indicate that our HRL approach can learn a diversity of options and that it can enhance the performance of RL in continuous control tasks.

## 1 Introduction

Reinforcement learning (RL) has been successfully applied to a variety of tasks, including board games (Silver et al., 2016), robotic manipulation tasks (Levine et al., 2016), and video games (Mnih et al., 2015). Hierarchical reinforcement learning (HRL) is a type of RL that leverages the hierarchical structure of a given task by learning a hierarchical policy (Sutton et al., 1999; Dietterich, 2000). Past studies in this field have shown that HRL can solve challenging tasks in the video game domain (Vezhnevets et al., 2017; Bacon et al., 2017) and robotic manipulation (Daniel et al., 2016; Osa et al., 2018b). In HRL, lower-level policies, which are often referred to as *option policies*, learn different behavior/control patterns, and the upper-level policy, which is often referred to as the *gating policy*, learns to select option policies. Recent studies have developed HRL methods using deep learning (Goodfellow et al., 2016) and have shown that HRL can yield impressive performance for complex tasks (Bacon et al., 2017; Frans et al., 2018; Vezhnevets et al., 2017; Haarnoja et al., 2018a). However, identifying the hierarchical policy structure that yields efficient learning is not a trivial task, since the problem involves learning a sufficient variety of types of behavior to solve a given task.

In this study, we present an HRL method via the mutual information (MI) maximization with advantage-weighted importance, which we refer to as adInfoHRL. We formulate the problem of learning a latent variable in a hierarchical policy as one of learning discrete and interpretable repre-

sentations of states and actions. Ideally, each option policy should be located at separate modes of the advantage function. To estimate the latent variable that corresponds to modes of the advantage function, we introduce advantage-weighted importance weights. Our approach can be considered to divide the state-action space based on an information maximization criterion, and it learns option policies corresponding to each region of the state-action space. We derive adInfoHRL as an HRL method based on deterministic option policies that are trained based on an extension of the deterministic policy gradient (Silver et al., 2014; Fujimoto et al., 2018). The contributions of this paper are twofold:

1. We propose the learning of a latent variable of a hierarchical policy as a discrete and hidden representation of the state-action space. To learn option policies that correspond to the modes of the advantage function, we introduce advantage-weighted importance.

2. We propose an HRL method, where the option policies are optimized based on the deterministic policy gradient and the gating policy selects the option that maximizes the expected return. The experimental results show that our proposed method adInfoHRL can learn a diversity of options on continuous control tasks. Moreover, our approach can improve the performance of TD3 on such tasks as the Walker2d and Ant tasks in OpenAI Gym with MuJoco simulator.

## 2 BACKGROUND

In this section, we formulate the problem of HRL in this paper and describe methods related to our proposal.

### 2.1 HIERARCHICAL REINFORCEMENT LEARNING

We consider tasks that can be modeled as a Markov decision process (MDP), consisting of a state space $\mathcal{S}$, an action space $\mathcal{A}$, a reward function $r : \mathcal{S} \times \mathcal{A} \mapsto \mathbb{R}$, an initial state distribution $\rho(\boldsymbol{s}_0)$, and a transition probability $p(\boldsymbol{s}_{t+1}|\boldsymbol{s}_t, \boldsymbol{a}_t)$ that defines the probability of transitioning from state $\boldsymbol{s}_t$ and action $\boldsymbol{a}_t$ at time $t$ to next state $\boldsymbol{s}_{t+1}$. The return is defined as $R_t = \sum_{i=t}^{T} \gamma^{i-t} r(\boldsymbol{s}_i, \boldsymbol{a}_i)$, where $\gamma$ is a discount factor, and policy $\pi(\boldsymbol{a}|\boldsymbol{s})$ is defined as the density of action $\boldsymbol{a}$ given state $\boldsymbol{s}$. Let $d^\pi(\boldsymbol{s}) = \sum_{t=0}^{T} \gamma^t p(\boldsymbol{s}_t = \boldsymbol{s})$ denote the discounted visitation frequency induced by the policy $\pi$. The goal of reinforcement learning is to learn a policy that maximizes the expected return $J(\pi) = \mathbb{E}_{\boldsymbol{s}_0, \boldsymbol{a}_0, \dots}[R_0]$ where $\boldsymbol{s}_0 \sim \rho(\boldsymbol{s}_0), \boldsymbol{a} \sim \pi$ and $\boldsymbol{s}_{t+1} \sim p(\boldsymbol{s}_{t+1}|\boldsymbol{s}_t, \boldsymbol{a}_t)$. By defining the Q-function as $Q^\pi(\boldsymbol{s}, \boldsymbol{a}) = \mathbb{E}_{\boldsymbol{s}_0, \boldsymbol{a}_0, \dots}[R_t|\boldsymbol{s}_t = \boldsymbol{s}, \boldsymbol{a}_t = \boldsymbol{a}]$, the objective function of reinforcement learning can be rewritten as follows:

$$J(\pi) = \iint d^\pi(\boldsymbol{s}) \pi(\boldsymbol{a}|\boldsymbol{s}) Q^\pi(\boldsymbol{s}, \boldsymbol{a}) \mathrm{d}\boldsymbol{a}\mathrm{d}\boldsymbol{s}. \tag{1}$$

Herein, we consider hierarchical policy $\pi(\boldsymbol{a}|\boldsymbol{s}) = \sum_{o \in \mathcal{O}} \pi(o|\boldsymbol{s}) \pi(\boldsymbol{a}|\boldsymbol{s}, o)$, where $o$ is the latent variable and $\mathcal{O}$ is the set of possible values of $o$. Many existing HRL methods employ a policy structure of this form (Frans et al., 2018; Vezhnevets et al., 2017; Bacon et al., 2017; Florensa et al., 2017; Daniel et al., 2016). In general, latent variable $o$ can be discrete (Frans et al., 2018; Bacon et al., 2017; Florensa et al., 2017; Daniel et al., 2016; Osa & Sugiyama, 2018) or continuous (Vezhnevets et al., 2017). $\pi(o|\boldsymbol{s})$ is often referred to as a *gating policy* (Daniel et al., 2016; Osa & Sugiyama, 2018), *policy over options* (Bacon et al., 2017), or *manager* (Vezhnevets et al., 2017). Likewise, $\pi(\boldsymbol{a}|\boldsymbol{s}, o)$ is often referred to as an *option policy* (Osa & Sugiyama, 2018), *sub-policy* (Daniel et al., 2016), or *worker* (Vezhnevets et al., 2017). In HRL, the objective function is given by

$$J(\pi) = \iint d^\pi(\boldsymbol{s}) \sum_{o \in \mathcal{O}} \pi(o|\boldsymbol{s}) \pi(\boldsymbol{a}|\boldsymbol{s}, o) Q^\pi(\boldsymbol{s}, \boldsymbol{a}) \mathrm{d}\boldsymbol{a}\mathrm{d}\boldsymbol{s}. \tag{2}$$

As discussed in the literature on inverse RL (Ziebart, 2010), multiple policies can yield equivalent expected returns. This indicates that there exist multiple solutions to latent variable $o$ that maximizes the expected return. To obtain the preferable solution for $o$, we need to impose additional constraints in HRL. Although prior work has employed regularizers (Bacon et al., 2017) and constraints (Daniel et al., 2016) to obtain various option policies, the method of learning a *good* latent variable $o$ that improves sample-efficiency of the learning process remains unclear. In this study we propose the learning of the latent variable by maximizing MI between latent variables and state-action pairs.

## 2.2 Deterministic Policy Gradient

The deterministic policy gradient (DPG) algorithm was developed for learning a monolithic deterministic policy $\boldsymbol{\mu_\theta}(\boldsymbol{s}) : \mathcal{S} \mapsto \mathcal{A}$ by Silver et al. (2014). In off-policy RL, the objective is to maximize the expectation of the return, averaged over the state distribution induced by a behavior policy $\beta(\boldsymbol{a}|\boldsymbol{s})$:

$$J(\pi) = \iint d^\beta(\boldsymbol{s})\pi(\boldsymbol{a}|\boldsymbol{s})Q^\pi(\boldsymbol{s}, \boldsymbol{a}) \mathrm{d}\boldsymbol{a}\mathrm{d}\boldsymbol{s}. \tag{3}$$

When a policy is deterministic, the objective becomes $J(\pi) = \int d^\beta(\boldsymbol{s})Q^\pi(\boldsymbol{s}, \boldsymbol{\mu_\theta}(\boldsymbol{s}))\mathrm{d}\boldsymbol{s}$. Silver et al. (2014) have shown that the gradient of a deterministic policy is given by

$$\nabla_{\boldsymbol\theta}\mathbb{E}_{\boldsymbol{s}\sim d^\beta(\boldsymbol{s})}[Q^\pi(\boldsymbol{s}, \boldsymbol{a})] = \mathbb{E}_{\boldsymbol{s}\sim d^\beta(\boldsymbol{s})}\left[\nabla_{\boldsymbol\theta}\boldsymbol{\mu_\theta}(\boldsymbol{s})\nabla_{\boldsymbol a}Q^\pi(\boldsymbol{s}, \boldsymbol{a})|_{\boldsymbol{a}=\boldsymbol{\mu_\theta}(\boldsymbol{s})}\right]. \tag{4}$$

The DPG algorithm has been extended to the deep deterministic policy gradient (DDPG) for continuous control problems that require neural network policies (Lillicrap et al., 2016). Twin Delayed Deep Deterministic policy gradient algorithm (TD3) proposed by Fujimoto et al. (2018) is a variant of DDPG that outperforms the state-of-the-art on-policy methods such as TRPO (Schulman et al., 2017a) and PPO (Schulman et al., 2017b) in certain domains. We extend this deterministic policy gradient to learn a hierarchical policy.

## 2.3 Representation Learning via Information Maximization

Recent studies such as those by Chen et al. (2016); Hu et al. (2017); Li et al. (2017) have shown that an interpretable representation can be learned by maximizing MI. Given a dataset $X = (\boldsymbol{x}_1, ..., \boldsymbol{x}_n)$, regularized information maximization (RIM) proposed by Gomes et al. (2010) involves learning a conditional model $\hat{p}(y|\boldsymbol{x}; \boldsymbol\eta)$ with parameter vector $\boldsymbol\eta$ that predicts a label $y$. The objective of RIM is to minimize

$$\ell(\boldsymbol\eta) - \lambda I_{\boldsymbol\eta}(\boldsymbol{x}, y), \tag{5}$$

where $\ell(\boldsymbol\eta)$ is the regularization term, $I_{\boldsymbol\eta}(\boldsymbol{x}, y)$ is MI, and $\lambda$ is a coefficient. MI can be decomposed as $I_{\boldsymbol\eta}(\boldsymbol{x}, y) = H(y) - H(y|\boldsymbol{x})$ where $H(y)$ is entropy and $H(y|\boldsymbol{x})$ the conditional entropy. Increasing $H(y)$ conduces the label to be uniformly distributed, and decreasing $H(y|\boldsymbol{x})$ conduces to clear cluster assignments. Although RIM was originally developed for unsupervised clustering problems, the concept is applicable to various problems that require learning a hidden discrete representation. In this study, we formulate the problem of learning the latent variable $o$ of a hierarchical policy as one of learning a latent representation of the state-action space.

# 3 Learning Options via Advantage-Weighted Information Maximization

In this section, we propose a novel HRL method based on advantage-weighted information maximization. We first introduce the latent representation learning via advantage-weighted information maximization, and we then describe the HRL framework based on deterministic option policies.

## 3.1 Latent Representation Learning via Advantage-Weighted Information Maximization

Although prior work has often considered $H(o|\boldsymbol{s})$ or $I(\boldsymbol{s}, o)$, which results in a division of the state space, we are interested in using $I\big((\boldsymbol{s}, \boldsymbol{a}), o\big)$ for dividing the state-action space instead. A schematic sketch of our approach is shown in Figure 1. As shown in the left side of Figure 1, the advantage function often has multiple modes. Ideally, each option policies should correspond to separate modes of the advantage function. However, it is non-trivial to find the modes of the advantage function in practice. For this purpose, we reduce the problem of finding modes of the advantage function to that of finding the modes of the probability density of state action pairs.

We consider a policy based on the advantage function of the form

$$\pi_{\mathrm{Ad}}(\boldsymbol{a}|\boldsymbol{s}) = \frac{f\big(A^\pi(\boldsymbol{s}, \boldsymbol{a})\big)}{Z}, \tag{6}$$

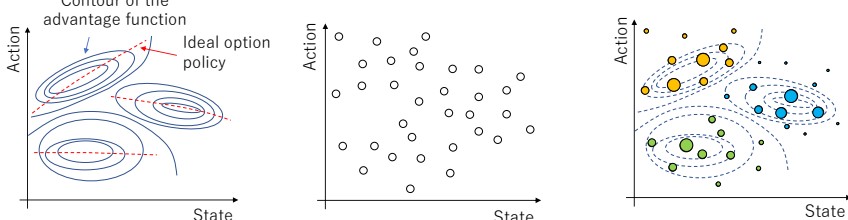

(a) Visualization of the advantage function with multiple modes in the state-action space.

(b) Modes of the density induced by arbitrary policies do not correspond to the modes of the advantage function.

(c) Modes of the density estimated with advantage-weighted importance correspond to the modes of the advantage function.

Figure 1: Schematic sketch of our HRL approach. By using the advantage-weighted importance, the problem of finding the modes of the advantage-function can be reduced to that of finding the modes of the density of state action pairs.

where $A^\pi(\boldsymbol{s}, \boldsymbol{a}) = Q^\pi(\boldsymbol{s}, \boldsymbol{a}) - V^\pi(\boldsymbol{s})$ is the advantage function, $V^\pi(\boldsymbol{s})$ is the state value function, and $Z$ is the partition function. $f(\cdot)$ is a *functional*, which is a function of a function. $f(\cdot)$ is a monotonically increasing function with respect to the input variable and always satisfies $f(\cdot) > 0$. In our implementation we used the exponential function $f(\cdot) = \exp(\cdot)$. When following such a policy, an action with the larger advantage is drawn with a higher probability. Under this assumption, finding the modes of the advantage function is equivalent to finding modes of the density induced by $\pi_{\text{Ad}}$. Thus, finding the modes of the advantage function can be reduced to the problem of clustering samples induced by $\pi_{\text{Ad}}$.

Following the formulation of RIM introduced in Section 2.3, we formulate the problem of clustering samples induced by $\pi_{\text{Ad}}$ as the learning of discrete representations via MI maximization. For this purpose, we consider a neural network that estimates $p(o|\boldsymbol{s}, \boldsymbol{a}; \boldsymbol{\eta})$ parameterized with vector $\boldsymbol{\eta}$, which we refer to as the *option network*. We formulate the learning of the latent variable $o$ as minimizing

$$\mathcal{L}_{\text{option}}(\boldsymbol{\eta}) = \ell(\boldsymbol{\eta}) - \lambda I\big(o, (\boldsymbol{s}, \boldsymbol{a}); \boldsymbol{\eta}\big), \tag{7}$$

where $I(o, (\boldsymbol{s}, \boldsymbol{a})) = \hat{H}(o|\boldsymbol{s}, \boldsymbol{a}; \boldsymbol{\eta}) - \hat{H}(o; \boldsymbol{\eta})$, and $\ell(\boldsymbol{\eta})$ is the regularization term. In practice, we need to approximate the advantage function, and we learn the discrete variable $o$ that corresponds to the modes of the current estimate of the advantage function. For regularization, we used a simplified version of virtual adversarial training (VAT) proposed by Miyato et al. (2016). Namely, we set $\ell(\boldsymbol{\eta}) = D_{\text{KL}}\big(p(o|\boldsymbol{s}^{\text{noise}}, \boldsymbol{a}^{\text{noise}}; \boldsymbol{\eta})||p(o|\boldsymbol{s}, \boldsymbol{a}; \boldsymbol{\eta})\big)$ where $\boldsymbol{s}^{\text{noise}} = \boldsymbol{s} + \boldsymbol{\epsilon_s}$, $\boldsymbol{a}^{\text{noise}} = \boldsymbol{a} + \boldsymbol{\epsilon_a}$, $\boldsymbol{\epsilon_s}$ and $\boldsymbol{\epsilon_a}$ denote white noise. This regularization term penalizes dissimilarity between an original state-action pair and a perturbed one, and Hu et al. (2017) empirically show that this regularization improves the performance of learning latent discrete representations.

When computing MI, we need to compute $p(o)$ and $H(o|\boldsymbol{s}, \boldsymbol{a})$ given by

$$p(o) = \int p^{\pi_{\text{Ad}}}(\boldsymbol{s}, \boldsymbol{a}) p(o|\boldsymbol{s}, \boldsymbol{a}; \boldsymbol{\eta}) \mathrm{d}\boldsymbol{a} \mathrm{d}\boldsymbol{s} = \mathbb{E}_{(\boldsymbol{s}, \boldsymbol{a}) \sim p^{\pi_{\text{Ad}}}(\boldsymbol{s}, \boldsymbol{a})} \left[ p(o|\boldsymbol{s}, \boldsymbol{a}; \boldsymbol{\eta}) \right] \tag{8}$$

$$H(o|\boldsymbol{s}, \boldsymbol{a}) = \mathbb{E}_{(\boldsymbol{s}, \boldsymbol{a}) \sim p^{\pi_{\text{Ad}}}(\boldsymbol{s}, \boldsymbol{a})} \left[ p(o|\boldsymbol{s}, \boldsymbol{a}; \boldsymbol{\eta}) \log p(o|\boldsymbol{s}, \boldsymbol{a}; \boldsymbol{\eta}) \right]. \tag{9}$$

Thus, the probability density of $(\boldsymbol{s}, \boldsymbol{a})$ induced by $\pi_{\text{Ad}}$ is necessary for computing MI for our purpose. To estimate the probability density of $(\boldsymbol{s}, \boldsymbol{a})$ induced by $\pi_{\text{Ad}}$, we introduce the advantage-weighted importance in the next section.

## 3.2 IMPORTANCE WEIGHTS FOR MUTUAL INFORMATION ESTIMATION

Although we show that the problem of finding the modes of the advantage function can be reduced to MI maximization with respect to the samples induced by $\pi_{\text{Ad}}$, samples induced by $\pi_{\text{Ad}}$ are not available in practice. While those induced during the learning process are available, a discrete representation obtained from such samples does not correspond to the modes of the advantage function. To estimate the density induced by $\pi_{\text{Ad}}$, we employ an importance sampling approach.

We assume that the change of the state distribution induced by the policy update is sufficiently small, namely, $d^{\pi_{\mathrm{Ad}}}(s) \approx d^{\beta}(s)$. Then, the importance weight can be approximated as

$$W(s, a) = \frac{p^{\pi_{\mathrm{Ad}}}(s, a)}{p^{\beta}(s, a)} = \frac{d^{\pi_{\mathrm{Ad}}}(s)\pi_{\mathrm{Ad}}(a|s)}{d^{\beta}(s)\beta(a|s)} \approx \frac{\pi_{\mathrm{Ad}}(a|s)}{\beta(a|s)} = \frac{f(A(s, a))}{Z\beta(a|s)}. \tag{10}$$

and the normalized importance weight is given gy

$$\tilde{W}(s, a) = \frac{W(s, a)}{\sum_{j=1}^{N} W(s_j, a_j)} = \frac{\frac{f(A(s,a))}{Z\beta(a|s)}}{\sum_{j=1}^{N} \frac{f(A(s_j, a_j))}{Z\beta(a_j|s_j)}} = \frac{\frac{f(A(s,a))}{\beta(a|s)}}{\sum_{j=1}^{N} \frac{f(A(s_j, a_j))}{\beta(a_j|s_j)}}. \tag{11}$$

As the partition function $Z$ is canceled, we do not need to compute $Z$ when computing the importance weight in practice. We call this importance weight $W$ the *advantage-weighted importance* and employ it to compute the objective function used to estimate the latent variable.

This advantage-weighted importance is used to compute the entropy terms for computing MI in Equation (7). The empirical estimate of the entropy $H(o)$ is given by

$$\hat{H}(o; \boldsymbol{\eta}) = -\sum_{o \in \mathcal{O}} \hat{p}(o; \boldsymbol{\eta}) \log \hat{p}(o; \boldsymbol{\eta}), \text{where } \hat{p}(o; \boldsymbol{\eta}) = \frac{1}{N} \sum_{i=1}^{N} W(s_i, a_i) p(o|s_i, a_i; \boldsymbol{\eta}). \tag{12}$$

where the samples $(s_i, a_i)$ are drawn from $p^{\beta}(s, a)$ induced by a behavior policy $\beta(a|s)$. Likewise, the empirical estimate of the conditional entropy $H(o|s, a)$ is given by

$$\hat{H}(o|s, a; \boldsymbol{\eta}) = \frac{1}{N} \sum_{i}^{N} W(s_i, a_i) p(o|s_i, a_i; \boldsymbol{\eta}) \log p(o|s_i, a_i; \boldsymbol{\eta}). \tag{13}$$

The derivations of Equations (12) and (13) are provided in Appendix A. To train the option network, we store the samples collected by the $M$ most recent behavior policies, to which we refer as on-policy buffer $D_{\mathrm{on}}$. Although the algorithm works with entire samples stored in the replay buffer, we observe that the use of the on-policy buffer for latent representation learning exhibits better performance. For this reason, we decided to use the on-policy buffer in our implementation. Therefore, while the algorithm is off-policy in the sense that the option is learned from samples collected by behavior policies, our implementation is "semi"on-policy in the sense that we use samples collected by the most recent behavior policies.

## 4 HRL OBJECTIVE WITH DETERMINISTIC OPTION POLICIES

Instead of stochastic option policies, we consider deterministic option policies and model them using separate neural networks. We denote by $\pi(a|s, o) = \boldsymbol{\mu}_{\boldsymbol{\theta}}^{o}(s)$ deterministic option policies parameterized by vector $\boldsymbol{\theta}$. The objective function of off-policy HRL with deterministic option policies can then be obtained by replacing $\pi(a|s)$ with $\sum_{o \in \mathcal{O}} \pi(o|s)\pi(a|s, o)$ in Equation (3):

$$J(\boldsymbol{w}, \boldsymbol{\theta}) = \int d^{\beta}(s) \sum_{o \in \mathcal{O}} \pi(o|s) Q^{\pi}(s, \boldsymbol{\mu}_{\boldsymbol{\theta}}^{o}(s); \boldsymbol{w}) ds, \tag{14}$$

where $Q^{\pi}(s, a; w)$ is an approximated Q-function parameterized using vector $w$. This form of the objective function is analogous to Equation (3). Thus, we can extend standard RL techniques to the learning of the gating policy $\pi(o|s)$ in HRL with deterministic option policies.

In HRL, the goal of the gating policy is to generate a value of $o$ that maximizes the conditional expectation of the return:

$$Q_{\Omega}^{\pi}(s, o) = \mathbb{E}[R|s_t = s, o_t = o] = \int \pi(a|s, o) Q^{\pi}(s, a) da, \tag{15}$$

which is often referred to as the option-value function (Sutton et al., 1999). When option policies are stochastic, it is often necessary to approximate the option-value function $Q_{\Omega}^{\pi}(s, o)$ in addition to the action-value function $Q^{\pi}(s, a)$. However, in our case, the option-value function for deterministic option policies is given by

$$Q_{\Omega}^{\pi}(s, o) = Q^{\pi}(s, \boldsymbol{\mu}_{\boldsymbol{\theta}}^{o}(s)), \tag{16}$$

---

**Algorithm 1** HRL via Advantage-Weighted Information Maximization (adInfoHRL)

---

**Input:** Number of options $O$, size of on-policy buffer
**Initialize:** Replay buffer $\mathcal{D}_R$, on-policy buffer $\mathcal{D}_{\text{on}}$, network parameters $\boldsymbol{\eta}$, $\boldsymbol{\theta}$, $\boldsymbol{w}$, $\boldsymbol{\theta}^{\text{target}}$, $\boldsymbol{w}^{\text{target}}$
**repeat**
    **for** $t = 0$ to $t = T$ **do**
        Draw an option for a given $\boldsymbol{s}$ by following Equation 17: $o \sim \pi(o|\boldsymbol{s})$
        Draw an action $\boldsymbol{a} \sim \beta(\boldsymbol{a}|\boldsymbol{s}, o) = \boldsymbol{\mu}_{\boldsymbol{\theta}}^o(\boldsymbol{s}) + \boldsymbol{\epsilon}$
        Record a data sample $(\boldsymbol{s}, \boldsymbol{a}, r, \boldsymbol{s}')$
        Aggregate the data in $\mathcal{D}_R$ and $\mathcal{D}_{\text{on}}$
        **if** the on-policy buffer is full **then**
            Update the option network by minimizing Equation (7) for samples in $\mathcal{D}_{\text{on}}$
            Clear the on-policy buffer $\mathcal{D}_{\text{on}}$
        **end if**
        Sample a batch $\mathcal{D}_{\text{batch}} \in \mathcal{D}_R$
        Update the Q network parameter $\boldsymbol{w}$
        **if** t mod d **then**
            Estimate $p(o|\boldsymbol{s}_i, \boldsymbol{a}_i)$ for $(\boldsymbol{s}_i, \boldsymbol{a}_i) \in \mathcal{D}_{\text{batch}}$ using the option network
            Assign samples $(\boldsymbol{s}_i, \boldsymbol{a}_i) \in \mathcal{D}_{\text{batch}}$ to the option $o^* = \arg \max p(o|\boldsymbol{s}_i, \boldsymbol{a}_i)$
            Update the option policy networks $\boldsymbol{\mu}_{\boldsymbol{\theta}}^o(\boldsymbol{s})$ for $o = 1, ..., O$ with Equation (19)
            Update the target networks: $\boldsymbol{w}_{\text{target}} \leftarrow \tau\boldsymbol{w} + (1-\tau)\boldsymbol{w}_{\text{target}}$, $\boldsymbol{\theta}_{\text{target}} \leftarrow \tau\boldsymbol{\theta} + (1-\tau)\boldsymbol{\theta}_{\text{target}}$
        **end if**
    **end for**
**until** the convergence
**return** $\boldsymbol{\theta}$

---

which we can estimate using the deterministic option policy $\boldsymbol{\mu}_{\boldsymbol{\theta}}^o(\boldsymbol{s})$ and the approximated action-value function $Q^\pi(\boldsymbol{s}, \boldsymbol{a}; \boldsymbol{w})$. In this work we employ the softmax gating policy of the form

$$\pi(o|\boldsymbol{s}) = \frac{\exp\left(Q^\pi(\boldsymbol{s}, \boldsymbol{\mu}_{\boldsymbol{\theta}}^o(\boldsymbol{s}); \boldsymbol{w})\right)}{\sum_{o\in\mathcal{O}} \exp\left(Q^\pi(\boldsymbol{s}, \boldsymbol{\mu}_{\boldsymbol{\theta}}^o(\boldsymbol{s}); \boldsymbol{w})\right)}, \tag{17}$$

which encodes the exploration in its form (Daniel et al., 2016). The state value function is given as

$$V^\pi(\boldsymbol{s}) = \sum_{o\in\mathcal{O}} \pi(o|\boldsymbol{s})Q^\pi(\boldsymbol{s}, \boldsymbol{\mu}_{\boldsymbol{\theta}}^o(\boldsymbol{s}); \boldsymbol{w}), \tag{18}$$

which can be computed using Equation (17). We use this state-value function when computing the advantage-weighted importance as $A(\boldsymbol{s}, \boldsymbol{a}) = Q(\boldsymbol{s}, \boldsymbol{a}) - V(\boldsymbol{s})$. In this study, the Q-function is trained in a manner proposed by Fujimoto et al. (2018). Two neural networks $(Q_{\boldsymbol{w}_1}^\pi, Q_{\boldsymbol{w}_2}^\pi)$ are trained to estimate the Q-function, and the target value of the Q-function is computed as $y_i = r_i + \gamma \min_{1,2} Q(\boldsymbol{s}_i, \boldsymbol{a}_i)$ for sample $(\boldsymbol{s}_i, \boldsymbol{a}_i, \boldsymbol{a}_i', r_i)$ in a batch sampled from a replay buffer, where $r_i = r(\boldsymbol{s}_i, \boldsymbol{a}_i)$. In this study, the gating policy determines the option once every $N$ time steps, i.e., $t = 0, N, 2N, \ldots$

Neural networks that model $\boldsymbol{\mu}_{\boldsymbol{\theta}}^o(\boldsymbol{a}|\boldsymbol{s})$ for $o = 1, ..., O$, which we refer to as *option-policy networks*, are trained separately for each option. In the learning phase, $p(o|\boldsymbol{s}, \boldsymbol{a})$ is estimated by the option network. Then, samples are assigned to option $o^* = \arg \max_o p(o|\boldsymbol{s}, \boldsymbol{a}; \boldsymbol{\eta})$ and are used to update the option-policy network that corresponds to $o^*$. When performing a rollout, $o$ is drawn by following the gating policy in Equation (17), and an action is generated by the selected option-policy network.

Differentiating the objective function in Equation (14), we obtain the deterministic policy gradient of our option-policy $\boldsymbol{\mu}_{\boldsymbol{\theta}}^o(\boldsymbol{s})$ given by

$$\nabla_{\boldsymbol{\theta}} J(\boldsymbol{w}, \boldsymbol{\theta}) = \mathbb{E}_{\boldsymbol{s}\sim d^\beta(\boldsymbol{s})\pi(o|\boldsymbol{s})}\left[\nabla_{\boldsymbol{\theta}}\boldsymbol{\mu}_{\boldsymbol{\theta}}^o(\boldsymbol{s})\nabla_{\boldsymbol{a}}Q^\pi(\boldsymbol{s}, \boldsymbol{a})|_{\boldsymbol{a}=\boldsymbol{\mu}_{\boldsymbol{\theta}}^o(\boldsymbol{s})}\right]. \tag{19}$$

The procedure of adInfoHRL is summarized by Algorithm 1. As in TD3 (Fujimoto et al., 2018), we employed the soft update using a target value network and a target policy network.

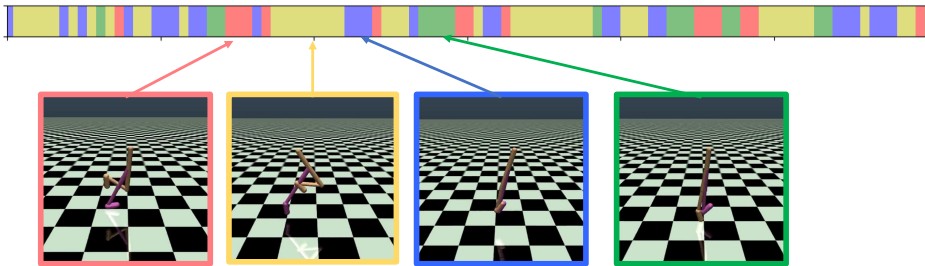

Figure 2: Activation of the four options over time steps on the Walker2d task.

## 5  EXPERIMENTS

We evaluated the proposed algorithm adInfoHRL on the OpenAI Gym platform (Brockman et al., 2016) with the MuJoCo Physics simulator (Todorov et al., 2012). We compared its performance with that of PPO implemented in OpenAI baselines (Dhariwal et al., 2017) and TD3. Henderson et al. (2018) have recently claimed that algorithm performance varies across environment, there is thus no clearly best method for all benchmark environments, and off-policy and on-policy methods have advantages in different problem domains. To analyze the performance of adInfoHRL, we compared it with state-of-the-art algorithms for both on-policy and off-policy methods, although we focused on the comparison with TD3, as our implementation of adInfoHRL is based on it. To determine the effect of learning the latent variable via information maximization, we used the same network architectures for the actor and critic in adInfoHRL and TD3. In addition, to evaluate the benefit of the advantage-weighted importance, we evaluated a variant of adInfoHRL, which does not use the advantage-weighted importance for computing mutual information. We refer to this variant of adInfoHRL as infoHRL. The gating policy updated variable $o$ once every three time steps. We tested the performance of adInfoHRL with two and four options.

The activation of options over time and snapshots of the learned option policies on the Walker2d task are shown in Figure 2, which visualizes the result from adInfoHRL with four options. One can see that the option policies are activated in different phases of locomotion. While the option indicated by yellow in Figure 2 corresponds to the phase for kicking the floor, the option indicated by blue corresponds to the phase when the agent was on the fly. Visualization of the options learned on the HalfCheetah and Ant tasks are shown in Appendix D.

The averaged return of five trials is reported in Figure 3(a)-(d). AdIfoHRL yields the best performance on Ant[1] and Walker2d, whereas the performance of TD3 and adInfoHRL was comparable on HalfCheetah and Hopper, and PPO outperformed the other methods on Hopper. Henderson et al. (2018) claimed that on-policy methods show their superiority on tasks with unstable dynamics, and our experimental results are in line with such previous studies. AdinfoHRL outperformed infoHRL, which isthe variant of adInfoHRL without the advantage-weighted importance on all the tasks. This result shows that the adavatage-weighted importance enhanced the performance of learning options.

AdInfoHRL exhibited the sample efficiency on Ant and Walker2d in the sense that it required fewer samples than TD3 to achieve comparable performance on those tasks. The concept underlying ad-InfoHRL is to divide the state-action space to deal with the multi-modal advantage function and learn option policies corresponding to separate modes of the advantage function. Therefore, adInfoHRL shows its superiority on tasks with the multi-modal advantage function and not on tasks with a simple advantage function. Thus, it is natural that the benefit of adInfoHRL is dependent on the characteristics of the task.

The outputs of the option network and the activation of options on Walker2d are shown in Figure 3(e)-(f), which visualize the result from adInfoHRL with four options. For visualization, the dimensionality was reduced using t-SNE (van der Maaten & Hinton, 2008). The state-action space

---

[1]We report the result on the Ant task implemented in rllab (Duan et al., 2016) instead of Ant-v1 implemented in the OpenAI gym, since the Ant task in the rllab is known to be harder than the Ant-v1 in the OpenAI gym. Results on Ant-v1 in the OpenAI gym is reported in Appendix D.

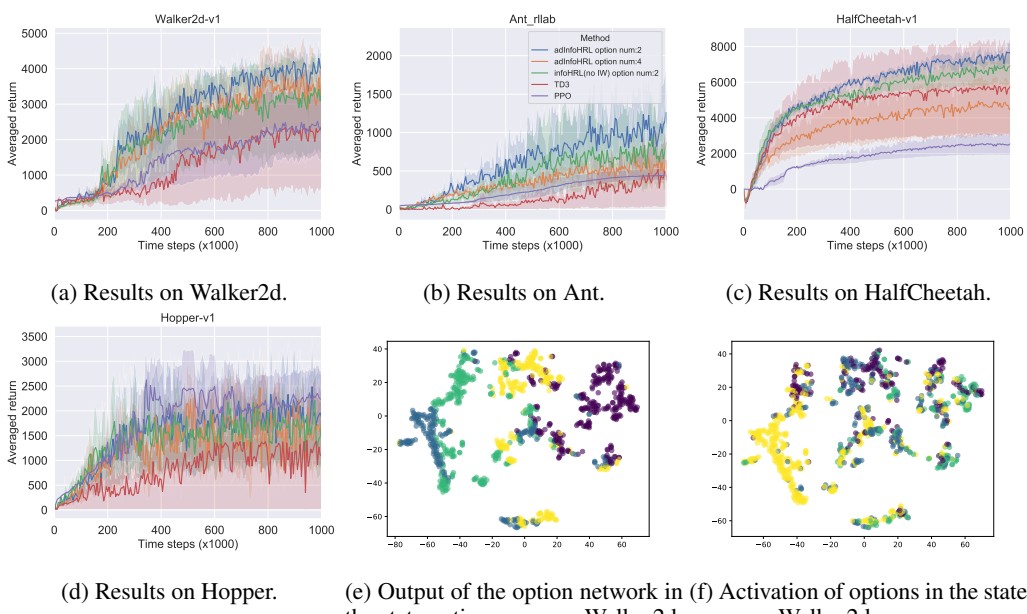

(a) Results on Walker2d.

(b) Results on Ant.

(c) Results on HalfCheetah.

(d) Results on Hopper.

(e) Output of the option network in the state-action space on Walker2d.

(f) Activation of options in the state space on Walker2d.

Figure 3: Performance of adInfoHRL. (a)-(d) show comparison with baseline methods. (e) and (f) show the output of the option network and the activation of options on Walker2d, respectively.

is clearly divided into separate domains in Figure 3(e). As shown in Figure 3(f), the options are activated in different domains of the state space, which indicates that diverse options are learned by adInfoHRL.

# 6 RELATED WORK AND DISCUSSION

Past studies have proposed several ways to deal with the latent variable in HRL. The recent work by Smith et al. (2018) proposed inferred option policy gradients (IOPG), which is derived as an extension of policy gradient to the option framework. Nachum et al. (2018) recently proposed off-policy target correction for HRL on goal-oriented tasks, where a higher-level policy instructs a lower-level policy by generating the goal signal instead of an inferred latent variable. A popular approach for learning the latent variable in HRL is the variational approach. The recent work by Haarnoja et al. (2018a) is based on soft actor critic (Haarnoja et al., 2018b), and the latent variable is inferred using the variational approach. The work by Hausman et al. (2018) is also closely related to the variational approach, and they proposed a method for learning a latent variable of a hierarchical policy via a variational bound. On the contrary, our method learns the latent variable by maximizing MI with advantage-weighted importance. Recent studies by Gregor et al. (2016); Florensa et al. (2017); Eysenbach et al. (2018) also considered the MI in their formulation. In these methods, MI between the state and the latent variable is considered so as to obtain diverse behaviors. Our approach is different from the previous studies in the sense that we employ MI between the latent variable and the state-action pairs, which leads to the division of the state-action space instead of considering only the state space. We think that dividing the state-action space is an efficient approach when the advantage function is multi-modal, as depicted in Figure 1. InfoGAIL proposed by Li et al. (2017) learns the interpretable representation of the state-action space via MI maximization. InfoGAIL can be interpreted as a method that divides the state-action space based on the density induced by an expert's policy by maximizing the regularized MI objective. In this sense, it is closely related to our method, although their problem setting is imitation learning (Osa et al., 2018a), which is different from our HRL problem setting.

The use of the importance weight based on the value function has appeared in previous studies (Dayan & Hinton, 1997; Kober & Peters, 2011; Neumann & Peters, 2009; Osa & Sugiyama, 2018). For example, the method proposed by Neumann & Peters (2009) employs the importance weight based on the advantage function for learning a monolithic policy, while our method uses a

similar importance weight for learning a latent variable of a hierarchical policy. Although Osa & Sugiyama (2018) proposed to learn a latent variable in HRL with importance sampling, their method is limited to episodic settings where only a single option is used in an episode.

Our method can be interpreted as an approach that divides the state-action space based on the MI criterion. This concept is related to that of Divide and Conquer (DnC) proposed by Ghosh et al. (2018), although DnC clusters the initial states and does not consider switching between option policies during the execution of a single trajectory.

In this study we developed adInfoHRL based on deterministic option policies. However, the concept of dividing the state-action space via advantage-weighted importance can be applied to stochastic policy gradients as well. Further investigation in this direction is necessary in future work.

## 7 CONCLUSIONS

We proposed a novel HRL method, hierarchical reinforcement learning via advantage-weighted information maximization. In our framework, the latent variable of a hierarchical policy is learned as a discrete latent representation of the state-action space. Our HRL framework is derived by considering deterministic option policies and by leveraging the analogy between the gating policy for HRL and a monolithic policy for the standard RL. The results of the experiments indicate that adInfoHRL can learn diverse options on continuous control tasks. Our results also suggested that our approach can improve the performance of TD3 in certain problem domains.

ACKNOWLEDGMENTS

MS was partially supported by KAKENHI 17H00757.

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

## A   MUTUAL INFORMATION WITH ADVANTAGE-WEIGHTED IMPORTANCE

The mutual information (MI) between the latent variable $o$ and the state action pair $(\boldsymbol{s}, \boldsymbol{a})$ is defined as

$$I\big((\boldsymbol{s}, \boldsymbol{a}), o\big) = H(o) - H(o|\boldsymbol{s}, \boldsymbol{a}) \tag{20}$$

where $H(o) = \int p(o) \log p(o) \mathrm{d}o$ and $H(o|\boldsymbol{s}, \boldsymbol{a}) = \int p(o|\boldsymbol{s}, \boldsymbol{a}) \log p(o|\boldsymbol{s}, \boldsymbol{a}) \mathrm{d}o$. We make the empirical estimate of MI employed by Gomes et al. (2010); Hu et al. (2017) and modify it to employ the importance weight. The empirical estimate of MI with respect to the density induced by a policy $\pi$ is given by

$$\hat{I}(\boldsymbol{s}, \boldsymbol{a}; o) = \sum_{o \in \mathcal{O}} \hat{p}(o) \log \hat{p}(o) - \hat{H}(o|\boldsymbol{s}, \boldsymbol{a}). \tag{21}$$

We consider the case where we have samples collected by a behavior policy $\beta(\boldsymbol{s}|\boldsymbol{a})$ and need to estimate MI with respect to the density induced by policy $\pi$. Given a model $p(o|\boldsymbol{s}, \boldsymbol{a}; \boldsymbol{\eta})$ parameterized by vector $\boldsymbol{\eta}$, $p(o)$ can be rewritten as

$$p(o) = \int p^\beta(\boldsymbol{s}, \boldsymbol{a}) \frac{p^\pi(\boldsymbol{s}, \boldsymbol{a})}{p^\beta(\boldsymbol{s}, \boldsymbol{a})} p(o|\boldsymbol{s}, \boldsymbol{a}; \boldsymbol{\eta}) \mathrm{d}\boldsymbol{a} \mathrm{d}\boldsymbol{s} = \mathbb{E}\left[W(\boldsymbol{s}, \boldsymbol{a}) p(o|\boldsymbol{s}, \boldsymbol{a}; \boldsymbol{\eta})\right], \tag{22}$$

where $W(\boldsymbol{s}, \boldsymbol{a}) = \frac{p^\pi(\boldsymbol{s}, \boldsymbol{a})}{p^\beta(\boldsymbol{s}, \boldsymbol{a})}$ is the importance weight. Therefore, the empirical estimate of $p(o)$ with respect to the density induced by a policy $\pi$ is given by

$$\hat{p}(o) = \frac{1}{N} \sum_{i=1}^N \tilde{W}(\boldsymbol{s}_i, \boldsymbol{a}_i) p(o|\boldsymbol{s}_i, \boldsymbol{a}_i; \boldsymbol{\eta}), \tag{23}$$

where $\tilde{W}(\boldsymbol{s}, \boldsymbol{a}) = \frac{\tilde{W}(\boldsymbol{s}, \boldsymbol{a})}{\sum_{j=1}^N \tilde{W}(\boldsymbol{s}_j, \boldsymbol{a}_j)}$ is the normalized importance weight.

Likewise, the conditional entropy with respect to the density induced by a policy $\pi$ is given by

$$H(o|\boldsymbol{s}, \boldsymbol{a}) = \int p^\pi(\boldsymbol{s}, \boldsymbol{a}) p(o|\boldsymbol{s}, \boldsymbol{a}; \boldsymbol{\eta}) \log p(o|\boldsymbol{s}, \boldsymbol{a}; \boldsymbol{\eta}) \mathrm{d}\boldsymbol{s} \mathrm{d}\boldsymbol{a} \tag{24}$$

$$= \int p^\beta(\boldsymbol{s}, \boldsymbol{a}) \frac{p^\pi(\boldsymbol{s}, \boldsymbol{a})}{p^\beta(\boldsymbol{s}, \boldsymbol{a})} p(o|\boldsymbol{s}, \boldsymbol{a}; \boldsymbol{\eta}) \log p(o|\boldsymbol{s}, \boldsymbol{a}; \boldsymbol{\eta}) \mathrm{d}\boldsymbol{s} \mathrm{d}\boldsymbol{a} \tag{25}$$

$$= \mathbb{E}\left[W(\boldsymbol{s}, \boldsymbol{a}) p(o|\boldsymbol{s}, \boldsymbol{a}; \boldsymbol{\eta}) \log p(o|\boldsymbol{s}, \boldsymbol{a}; \boldsymbol{\eta})\right]. \tag{26}$$

Therefore, the empirical estimate of the conditional entropy with respect to the density induced by a policy $\pi$ is given by

$$\hat{H}(o|\boldsymbol{s}, \boldsymbol{a}) = \frac{1}{N} \sum_{i=1}^N W(\boldsymbol{s}_i, \boldsymbol{a}_i) p(o|\boldsymbol{s}_i, \boldsymbol{a}_i; \boldsymbol{\eta}) \log p(o|\boldsymbol{s}_i, \boldsymbol{a}_i; \boldsymbol{\eta}). \tag{27}$$

Thus, the empirical estimates of MI can be computed by Equations (21), (23) and (27).

## B   DERIVATION OF THE STATE-VALUE FUNCTION

In HRL, the value function is given by

$$V(\boldsymbol{s}) = \int \sum_{o \in \mathcal{O}} \pi(o|\boldsymbol{s}) \pi(\boldsymbol{a}|\boldsymbol{s}, o) Q^\pi(\boldsymbol{s}, \boldsymbol{a}) \mathrm{d}\boldsymbol{a} = \sum_{o \in \mathcal{O}} \pi(o|\boldsymbol{s}) \int \pi(\boldsymbol{a}|\boldsymbol{s}, o) Q^\pi(\boldsymbol{s}, \boldsymbol{a}) \mathrm{d}\boldsymbol{a} \tag{28}$$

Since option policies are deterministic given by $\boldsymbol{\mu}_{\boldsymbol{\theta}}^o(\boldsymbol{s})$, the state-value function is given by

$$V(\boldsymbol{s}) = \sum_{o \in \mathcal{O}} \pi(o|\boldsymbol{s}) Q^\pi(\boldsymbol{s}, \boldsymbol{\mu}_{\boldsymbol{\theta}}^o(\boldsymbol{s})) \mathrm{d}\boldsymbol{a}. \tag{29}$$

Table 1: Hyperparameters of adInfoHRL used in the experiment.

| Description | Symbol | Value |
|---|---|---|
| Coefficient for updating the target network | $\tau$ | 0.005 |
| Discount factor | $\gamma$ | 0.99 |
| Learning rate for actor | | 0.001 |
| Learning rate for critic | | 0.001 |
| Batch size for critic | | 100 |
| Total batch size for all option policies | | 200 (option num=2), 400 (option num=4) |
| Batch size for the option network | | 50 |
| Size of the on-policy buffer | | 5000 |
| Number of epochs for training the option network | | 40 |
| Number of units in hidden layers | | (400, 300) |
| Activation function | | Relu, Relu, tanh |
| optimizer | | Adam |
| noise clip threshold | c | 0.5 |
| noise for exploration | | 0.1 |
| action noise for the critic update | | 0.2 |
| variance of the noise for MI regularization | | 0.04 |
| coefficient for the MI term | $\lambda$ | 0.1 |

Table 2: Hyperparameters of TD3 used in the experiment.

| Description | Symbol | Value |
|---|---|---|
| Coefficient for updating the target network | $\tau$ | 0.005 |
| Discount factor | $\gamma$ | 0.99 |
| Learning rate for actor | | 0.001 |
| Learning rate for critic | | 0.001 |
| Batch size | | 100 |
| Number of units in hidden layers | | (400, 300) |
| Activation function | | Relu, Relu, tanh |
| optimizer | | Adam |
| noise clip threshold | c | 0.5 |
| noise for exploration | | 0.1 |
| action noise for the critic update | | 0.2 |

## C  EXPERIMENTAL DETAILS

We performed evaluations using benchmark tasks in the OpenAI Gym platform (Brockman et al., 2016) with Mujoco physics simulator (Todorov et al., 2012). Hyperparameters of reinforcement learning methods used in the experiment are shown in Tables 1-3. For exploration, both adInfoHRL and TD3 used the clipped noise drawn from the normal distribution as $\epsilon \sim \text{clip}(\mathcal{N}(0, \sigma), -c, c)$, where $\sigma = 0.2$ and $c = 0.5$. For hyperparameters of PPO, we used the default values in OpenAI baselines (Dhariwal et al., 2017). For the Walker2d, HalfCheetah, and Hopper tasks, we used the Walker2d-v1, HalfCHeetah-v1, and Hopper-v1 in the OpenAI Gym, respectively. For the Ant task, we used the AntEnv implemented in the rllab (Duan et al., 2016). When training a policy with AdInfoHRL, infoHRL, and TD3, critics are trained once per time step, and actors are trained once every after two updates of the critics. The source code is available at `https://github.com/TakaOsa/adInfoHRL`.

We performed the experiments five times with different seeds, and reported the averaged test return where the test return was computed once every 5000 time steps by executing 10 episodes without exploration. When performing the learned policy without exploration, the option was drawn as

$$o = \max_{o'} Q^\pi(\boldsymbol{s}, \boldsymbol{\mu}^{o'}(\boldsymbol{s})), \tag{30}$$

instead of following the stochastic gating policy in Equations (17).

Table 3: Hyperparameters of PPO used in the experiment. We tuned hyperparameters for our tasks, which are defferent from the default parameters in OpenAI baselines (Dhariwal et al., 2017).

| Description | Symbol | Value |
|---|---|---|
| Coefficient for updating the target network | $\tau$ | 0.001 |
| Discount factor | $\gamma$ | 0.99 |
| Batch size | | 2048 |
| Number of units in hidden layers | | (64, 64) |
| Clipping parameter | $\epsilon$ | 0.15 |
| Initial learning rate | | 0.0005 |
| Learning rate schedule | | linear |

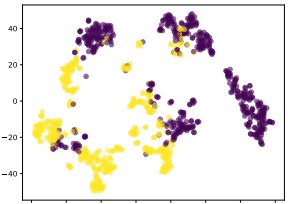

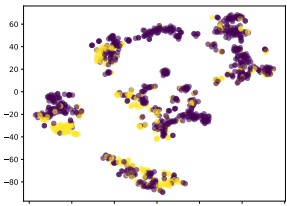

(a) Output of the option network in the state-action space on HalfCheetah-v1.

(b) Activation of options in the state space on HalfCheetah-v1.

Figure 4: Distribution of options on the HalfCheetah-v1 task using adInfoHRL with two options. The dimensionality is reduced by t-SNE for visualization.

## D    ADDITIONAL INFORMATION ON EXPERIMENTAL RESULTS

On the HalfCheetah task, adInfoHRL delivered the best performance with two options. The distribution of options on HalfCheetah0v1 after one million steps is shown in Figure 4. Although the state-action space is evenly divided, the options are not evenly activated. This behavior can occur because the state-action space is divided based on the density induced by the behavior policy while the activation of options is determined based on the quality of the option policies in a given state. Moreover, an even division in the action-state space is not necessarily the even division in the state space.

The activation of the options over time is shown in Figure 5. It is clear that one of the option corresponds to the stable running phase and the other corresponds to the phase for recovering from unstable states.

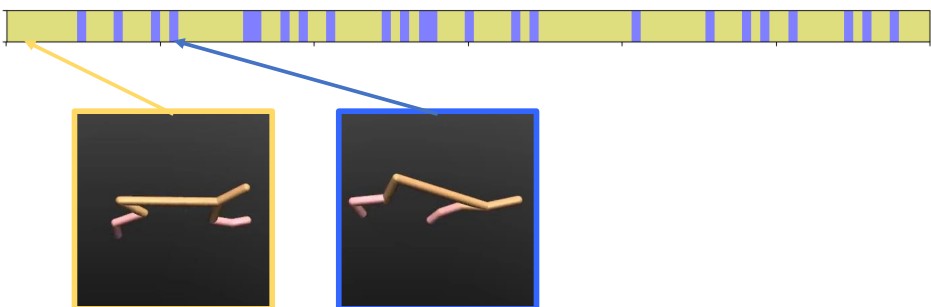

Figure 5: Activation of options over time steps on the HalfCheetah-v1 task using adInfoHRL with two options.

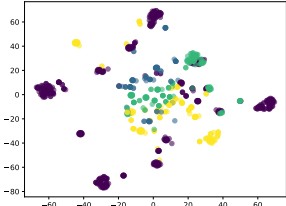 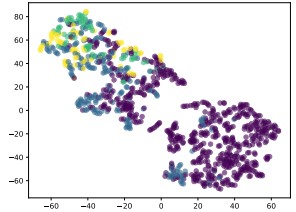

(a) Output of the option network in the state-action space on the Ant-rllab task.

(b) Activation of options in the state space on the Ant-rllab task.

Figure 6: Distribution of options on Ant-rllab task using adInfoHRL with four options. The dimensionality is reduced by t-SNE for visualization.

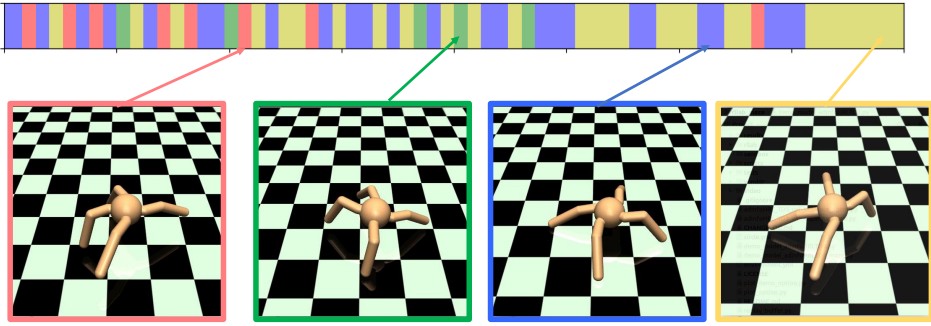

Figure 7: Activation of the options over time steps on Ant-rllab task. Four options are learned.

The distribution of four options on the Ant-rllab task after one million steps is shown in Figure 6. Four options are activated in the different domains of the state space. The activation of the options over time on the Ant-rllab task is shown in Figure 7. While four options are actively used in the beginning of the episode, two (blue and yellow) options are mainly activated during the stable locomotion.

Since the Ant task implemented in rllab is known to be harder than the Ant-v1 implemented in the OpenAI gym, we reported the result of the Ant task in rllab in the main manuscript. Here, we report the result of the Ant-v1 task implemented in the OpenAI gym. On the Ant-v1 task, adInfoHRL yielded the best performance with two options. The performance of adInfoHRL with two options is comparable to that of TD3 on Ant-v1. This result indicates that the Ant-v1 task does not require a hierarchical policy structure, while a hierarchical policy improves the performance of learning on Ant-rllab. The distribution of options on Ant-v1 task after one million steps is shown in Figure 8. The activation of the options over time is shown in Figure 9. It is evident that two option policies on the Ant-v1 task corresponded to different postures of the agent.

A recent study on HRL by Smith et al. (2018) reported the performance of IOPG on Walker2d-v1, Hopper-v1, and HalfCheetah-v1. The study by Haarnoja et al. (2018a) reported the performance of SAC-LSP on Walker2d-v1, Hopper-v1, HalfCheetah-v1, and Ant-rllab. A comparison of performance between our method, IOPG, and SAC-LSP is summarized in Table 4. We report the performance after 1 million steps. It is worth noting that adInfoHRL outperformed IOPG on these tasks in terms of the achieved return, although we are aware that the qualitative performance is also important in HRL. AdInfoHRL outperformed SAC-LSP on Walker2d-v1 and Ant-rllab, and SAC-LSP shows its superiority on HalfCheetah-v1 and Hopper-v1. However, the results of SAC-LSP were obtained by using reward scaling, which was not used in the evaluation of adInfoHRL. Therefore, further experiments are necessary for fair comparison under the same condition.

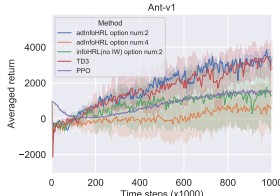 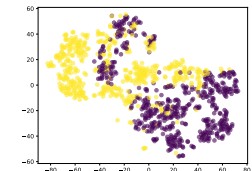 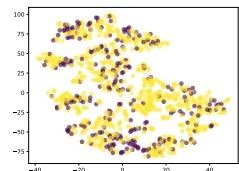

(a) Averaged return on the Ant-v1 task in the OpenAI gym. (b) Output of the option network in the state-action space on Ant-v1. (c) Activation of options in the state space on Ant-v1.

Figure 8: Distribution of options on the Ant-v1 task using adInfoHRL with two options. The dimensionality is reduced by t-SNE for visualization.

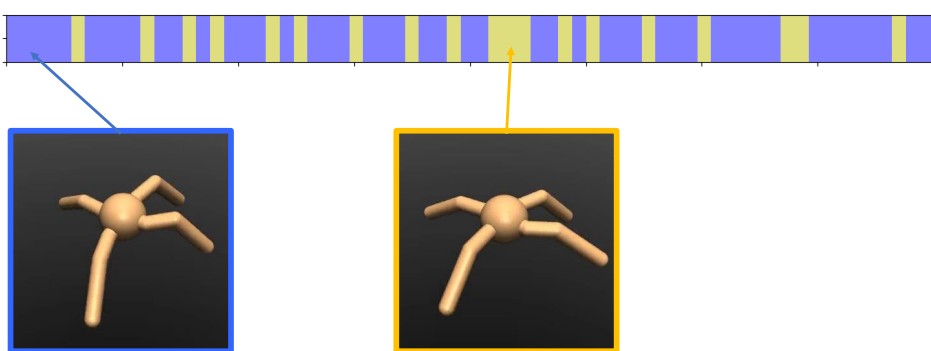

Figure 9: Activation of options over time steps on the Ant-v1 task using adInfoHRL with two options.

Table 4: Comparison of performance with existing methods after 1 million steps as reported in the literature. For adInfoHRL, we show the mean and the standard deviation of the results from the 10 different seeds. The performance of IOPG and SAC-LSP is from the original papers (Smith et al., 2018) and (Haarnoja et al., 2018a)

| task | adInfoHRL (two opt.) | adInfoHRL (four opt.) | IOPG | SAC-LSP |
|---|---|---|---|---|
| Walker2d-v1 | **3752.1 ± 442** | 3404.2± 785.6 | ≈ 800 | ≈ 3000 |
| HalfCheetah-v1 | 6315.1 ± 612.8 | 4520.6 ± 859.3 | ≈ 800 | **≈ 8000** |
| Hopper-v1 | 1821.7 ± 626.3 | 1717.5 ± 281.7 | ≈ 1000 | **≈ 2500** |
| Ant rllab | **1263.2 ± 333.5** | 683.2 ± 105.68 | – | ≈ 500 |

