# OpenReview forum: "Hierarchical Reinforcement Learning via Advantage-Weighted Information Maximization"
_ICLR.cc/2019/Conference_

### Official Review · AnonReviewer2 · 2018-11-02
**Interesting ideas and analysis but somewhat unclear motivation and limited empirical evaluation**

**Rating:** 7
**Confidence:** 4

**Review:**

Revision: The authors addressed most of my concerns and clearly put in effort to improve the paper. The paper explains the central idea better, is more precise in terminology in general, and the additional ablation gives more insight into the relative importance of the advantage weighting. I still think that the results are a bit limited in scope but the idea is interesting and seems to work for the tasks in the paper. I adjusted my score to reflect this.

Summary:
The paper proposes an HRL system in which the mutual information of the latent (option) variable and the state-action pairs is approximately maximized. To approximate the mutual information term, samples are reweighted based on their estimated advantage. TD3 is used to optimize the modules of the system. The system is evaluated on continuous control task from OpenAI gym and rllab.

For the most part, the paper is well-written and it provides a good overview of related work and relevant terminology. The experiments seem sound even though the results are not that impressive. The extra analysis of the option space and temporal distribution is interesting.

Some parts of the theoretical justification for the method are not entirely clear to me and would benefit from some clarification. Most importantly, it is not clear to me why the policy in Equation 7 is considered to be optimal. Given some value or advantage function, the optimal policy would be the one that picks the action that maximizes it. The authors refer to earlier work in which similar equations are used, but in those papers this is typically in the context of some entropy maximizing penalty or KL constraint. A temperature parameter would also influence the exploration-exploitation trade-off in this ‘optimal’ policy. I understand that the rough intuition is to take actions with higher advantage more often while still being stochastic and exploring but the motivation could be more precise given that most of the subsequent arguments are built on top of it. However, this is not the policy that is used to generate behavior. In short, the paper is clear enough about how the method is constructed but it is not very clear to me *why* the mutual information should be optimized with respect to this 'optimal' policy instead of the actual policy one is generating trajectories from.

HRL is an interesting area of research with the potential to learn complicated behaviors. However, it is currently not clear how to evaluate the importance/usefulness of hierarchical RL systems directly and the tasks in the paper are still solvable by standard systems. That said, the occasional increase in sample efficiency over plain TD3 looks promising. It is somewhat disappointing that the number of beneficial option is generally so low. To get more insight in the methods it would have been nice to see a more systematic ablation of related methods with different mutual information pairings (action or state only) and without the advantage weighting. Could it be that the number of options has to remain limited because there is no parameter sharing between them? It would be interesting to see results on more challenging control problems where the hypothesized multi-modal advantage structure is more likely to be present.

All in all I think that this is an interesting paper but the foundations of the theoretical motivation need a bit more clarification. In addition, experiments on more challenging problems and a more systematic comparison with similar models would make this a much stronger paper.

Minor issues/typos:
- Contributions 2 and 3 have a lot of overlap.
- The ‘o’ in Equation 2 should not be bold font.
- Appendix A. Shouldn’t there be summations over ‘o’ in the entropy definitions?

---

> ### Author Response · Authors · 2018-11-13
> **The reason why the advantage-weighted importance is necessary**
>
> Thank you for the comments. We revised our manuscript to clarify the motivation. Please refer to the above post for details. We would also like to clarify some points here.
>
> - The reason why the advantage-weighted importance is necessary
> If we do not use the advantage-weighted importance, we learn the latent variable with respect to the density of state-action pairs visited during the learning phase. However, modes of such a density correspond to not modes of the advantage function but the current location of the option policies. Therefore, the latent variable learned without the advantage-weighted importance do not improve the location of the option policies. By using the advantage-weighted importance, we can learn the discrete variable that corresponds to the modes of the advantage function.

---

> > ### Author Response · Authors · 2018-11-22
> > **The performance without advantage-weighted importance**
> >
> > To evaluate the benefit of the advantage-weighted importance, we evaluated a variant of adInfoHRL, which does not use the advantage-weighted importance for computing mutual information. The results show that the proposed method outperforms the version without the advantage-weighted importance on all the four tasks. We added the result of the results in the revised manuscript.

---

### Official Review · AnonReviewer3 · 2018-11-04
**Potentially interesting idea, but a very poorly written paper**

**Rating:** 6
**Confidence:** 4

**Review:**

The authors propose an HRL algorithm that attempts to learn options that maximize their mutual information with the state-action density under the optimal policy.

Several key terms are used in ways that differ from the rest of the literature. The authors claim options are learned in an "unsupervised" manner, but it is unclear what this means. Previous work (none of which is cited) has dealt with unsupervised option discovery in the context of mutual information maximization (Variational intrinsic control, diversity is all you need, etc), but they do so in the absence of reward, unlike this paper. "Optimal policy" is similarly abused, with it appearing to mean optimal from the perspective of the current model parameters, rather than optimal in any global sense. Or at least I think that is what the authors intend. If they do mean the globally optimal policy, then its unclear how to interpret Equation 8, with its reference to a behavior policy and an advantage function, neither of which would be available if meant to represent the global optimum.

Equation 10 comes out of nowhere. One must assume they meant "maximize mutual information" and not "minimize", but who knows. Why is white-noise being added to the states and actions? Is this some sort of noise-contrastive estimation approach to mutual information estimation? It doesn't appear to be, but it is unclear what else could motivate it. Even the appendices fail to shine light on this equation.

The algorithm block isn't terribly helpful. The "t" variable is used outside of its for loop, which draws into question the exact nesting structure of the underlying algorithm (which isn't obvious for HRL methods). There aren't any equations referenced, with the option policy network's update not even referencing the loss nor data over which the loss would be evaluated.

Some of the experimental results show promise, but the PPO Ant result raises some questions. Clearly the OpenAI implementation of PPO used would have tuned for the OpenAI gym Ant implementation, and the appendix shows it getting decent results. But it never takes off in the harder RlLab version -- were the hyper-parameters adjusted for this new environment?

It is also odd that no other HRL approaches are evaluated against, given the number cited. Running these methods might be too costly, but surely a table comparing results reported in those papers should be included.

A minor point: another good baseline would be TD3 with the action repeat adjusted to be inline with the gating policy.

I apologise if this review came off as too harsh -- I believe a good paper can be made of this with extensive rewrites and additional experiments. But the complete lack of clarity makes it feel like it was rushed out prematurely.

EDIT: Now this is a paper that makes sense! With the terminology cleared up and the algorithm fully unpacked, this approach seems quite interesting. The experimental results could always be stronger, but no longer have any holes in them. Score 3-->6

---

> ### Author Response · Authors · 2018-11-13
> **Clarification of the objective function for learning the latent variable**
>
> Thank you for the comments. We revised our manuscript to clarify the motivation. Please refer to the above post for details. We also answer your question here.
>
> -	Clarification of the objective function for learning the latent variable
> Reviewer 3 raised a concern on the objective function for learning the latent variable. The objective function is based on regularized information maximization (RIM) . Since the objective function is negative to the MI term, the latent variable is learned by minimizing the objective function. The KL term in the objective function is the regularization term based on virtual adversarial training (VAT). We revised our manuscript to make the story more easily followable.
>
> -	Hyperparameters of PPO
> We used the default parameters in the baseline implementation, and we did not tune the parameter for Ant-rllab. We fixed the hyperparameters of adInfoHRL and TD3 as well, and we did not tune hyperparameters for specific tasks. The hyperparmeters are provided in Appendix.

---

> > ### Comment · AnonReviewer3 · 2018-11-13
> > **PPO baseline**
> >
> > While you didn't tune hyperparameters for specific tasks, surely you picked hyperparameters by maximizing performance across all tasks. PPO's hyperparameters were tuned without knowledge of Ant-rllab, making the current comparison unfair. Rerunning a PPO hyperparameter sweep with your collection tasks would solve this issue, as would limiting the set of tasks to those used to tune PPO (i.e. switching out Ant-rllab for Ant-gym).

---

> > > ### Author Response · Authors · 2018-11-19
> > > **PPO baseline is updated**
> > >
> > > We performed a parameter sweep to tune the performance of PPO, and we updated the result graph. We observe that there is a trade-off of the performance across the tasks. For example,  when obtaining the better performance on Ant-rllab, the performance on the Walkder2d-v1 gets lower. We picked the hyperparameters of PPO that give the performance comparable to the one reported in [Haarnoja, ICML 2018],  although the hyperparameters of PPO used in [Haarnoja, ICML 2018] are not provided.

---

### Official Review · AnonReviewer1 · 2018-11-05
**Good idea, exposition can be much improved**

**Rating:** 5
**Confidence:** 4

**Review:**

The paper considers the problem of hierarchical reinforcement learning, and proposes a criterion that aims to maximize the mutual information between options and state-action pairs.

The idea of having options partition the state-action space is appealing, because this allows options visit the same states, so long as they act differently, which is natural. The authors show empirically that the learned options do indeed decompose the state-action space, but not the state space.

There is a lot in the paper already, but the exposition could be much improved. Many of the design choices appear very ad hoc, and some are outright confusing. Some detailed comments:

* I got really confused in Section 3 re: advantage-weighted importance sampling. Why do this? If the option policies are trying to optimize reward, won’t they become optimal eventually (or so we usually hope in RL)? This section seems to assume that the advantage function is somehow given. It also doesn’t look like this gets used in the actual algorithm, and in fact on page 5 it is stated that “we decided to use the on-policy buffer in our implementation”. Then why introduce the off-policy bit at all, and list it as a contribution?
* Please motivate the choices. The paper mentions that one of its contributions are options with deterministic policies. This isn’t a contribution unless it addresses some problem that stochastic policies fail at. For example, DPG allows one to address continuous control problems.
Same with using information maximization. The paper literally states that “an interpretable representation can be learned by maximizing mutual information”. Representation of what? MI between what?
* Although the qualitative results are nice (separation of the state-action space), empirical results are modest at best. This may be ok, because based on the partition of the state-action space it seems that the option policies learn diverse behaviors in the same states. Maybe videos visualizing different options from the same states would be informative.
* Please add more discussion on why the options are switched at every step

---

> ### Author Response · Authors · 2018-11-13
> **Our method learns discrete representations of the state-action space**
>
> Thank you for the comments. We answer some of your questions here. Please refer to the above post for other concerns and questions.
>
> - questions about information maximization
> Our approach is to maximize the mutual information between the latent variable of the hierarchical policy and the state-action pairs, which results in learning discrete representations of the state-action space. We revised the manuscript to clarify this point.
>
> - Please add more discussion on why the options are switched at every step
> The options are not switched at every time step as shown in Figure 2. For example, the option indicated by yellow is activated for about 30 time-steps at most.
>
> - Question about whether our method is off-policy or not
> We do not intend to list “off-policy” as one of the contributions, although it is one of the features of our approach. Our approach is off-policy in several points even though we employed an on-policy buffer for learning the options. In our method, samples are collected using a behavior policy instead of the “raw” learned policy, and both the Q-function and the option policies are trained using the replay buffer in an off-policy manner. Therefore, we think that our method should be categorized as an off-policy method.
>
> -	Availability of the advantage function
> We do not assume the availability of the advantage function. In practice, it is necessary to approximate the advantage function. Our approach finds the latent variable with respect to the current estimate of the advantage function. Since the Q-function converges to the optimum as learning progresses, our method can learn the latent variable with respect to the optimal advantage function at convergence. In actor critic, policy parameters are updated with respect to the current approximation of the Q-function or the advantage function. Likewise, one can interpret that the latent variable of our hierarchical policy is updated with respect to the current approximation of the advantage function in our method.

---

### Author Response · Authors · 2018-11-13
**summary of our revision and answers to reviewers’ questions**

Dear reviewers

Thank you for constructive comments. We made major revision, especially on the part where we motivate and explain our method. We believe that the manuscript is significantly improved thanks to the reviewers’ comments.

Here is the summary of our revision and answers to reviewers’ questions
1.	Removal of the term “optimal policy”
In the initial manuscript, a policy of the form \frac{\exp(A(s,a ))}{Z} is referred to as “optimal policy”, and we removed this expression. We consider a policy of this form in order to reduce the problem of finding the modes of the advantage function to that of finding modes of the probability density of state-action pairs. Any policy from which a sample is drawn and that results in a higher return with higher probability can be used for this purpose. In the revised manuscript, a policy of the form \frac{f(A(s,a ))}{Z} is referred to as “a policy based on the advantage function,”  where f is a monotonically increasing function with respect to the input variable. We replaced \exp with a monotonically increasing function f in the revised manuscript so that we can emphasize that the form of Equation 7 is not limited to the exponential function. Although we used f() = exp() in our implementation and a policy of the form \frac{\exp(A(s,a ))}{Z} is optimal in entropy-regularized RL, our method is not related to entropy-regularized RL. We revised the manuscript to avoid the confusion.

2.	Clarification of the motivation of using the advantage-weighted importance
We can reduce the problem of finding the modes of the advantage function to that of the modes of the density of state-action pairs with the advantage-weighted importance. However, without the advantage-weighted importance, modes of the density of the state-action pairs induced by an arbitrary policy do no correspond to those of the advantage function in general. We revised the manuscript to clarify this point.

3.	Benefit of the deterministic option policies
Reviewer 1 questioned the benefit of the deterministic option policies. When learning stochastic option policies, the option-value function needs to be learned in addition to the action-value functions. As discussed in Section 4 in the revised manuscript, the option-value function does not need to be learned, since it can be estimated from the action-value function and the option policies when the option policies are deterministic. When option policies are stochastic, learning the option-value function needs to be updated if the option policies are updated. However, in the case of deterministic option policies, this additional learning cost is not necessary. Hence, the use of deterministic option policies can be more sample-efficient than that of stochastic option policies.

4.	Comparison with other HRL methods
We put a table for comparison with recent HRL methods in Appendix. In terms of the achieved returns, our method outperforms IOPG (Smith et al., ICML 2018). Compared with SAC-LSP (Haanoja, ICML2018), our method outperforms SAC-LSP on Walker2d and Ant-rllab, and SAC-LSP shows its superiority on Hopper.

5.	Revision of premature descriptions
Reviewer 3 pointed out some issues of the description in Algorithm 1, and Reviewer 2 also pointed out some typos. We modified those points and revised several descriptions to improve the clarity. In addition, the term “unsupervised” was confusing in the initial manuscript, we removed the related descriptions. We also cited missing related work, such as variational intrinsic control and diversity is all you need.

---

> ### Author Response · Authors · 2018-11-22
> **Update log**
>
> - The PPO baseline is updated to address the concern from the reviewer 3
> - The experimental results are updated to include the performance of a variant of the proposed method which does not use the advantage-weighted importance for computing mutual information.

---

### Meta-Review · Area_Chair1 · 2018-12-14

**Confidence:** 4
**Recommendation:** Accept (Poster)

**Metareview:**

This paper proposes a method for hierarchical reinforcement learning that aims to maximize mutual information between options and state-action pairs. The approach and empirical analysis is interesting. The initial submission had many issues with clarity. However, the new revisions of the paper have significantly improved the clarity, better describing the idea and improving the terminology. The main remaining weakness is the scope of the experimental results.
However, the reviewers agree that the paper exceeds the bar for publication at ICLR with the existing experiments.